# Visualizing Oral Infection Dynamics of *Beauveria bassiana* in the Gut of *Tribolium castaneum*

**DOI:** 10.3390/jof11020101

**Published:** 2025-01-28

**Authors:** Lautaro Preisegger, Juan Cruz Flecha, Fiorella Ghilini, Daysi Espin-Sánchez, Eduardo Prieto, Héctor Oberti, Eduardo Abreo, Carla Huarte-Bonnet, Nicolás Pedrini, Maria Constanza Mannino

**Affiliations:** 1Instituto de Investigaciones Bioquímicas de La Plata (INIBIOLP), CCT La Plata Consejo Nacional de Investigaciones Científicas y Técnicas (CONICET), Universidad Nacional de La Plata (UNLP), Calles 60 y 120, La Plata 1900, Argentina; lpreisegger@med.unlp.edu.ar (L.P.); juancruzflecha16@gmail.com (J.C.F.); despinsanchez@med.unlp.edu.ar (D.E.-S.); carlahb@biol.unlp.edu.ar (C.H.-B.); 2Instituto de Investigaciones Fisicoquìmicas Teóricas y Aplicadas (INIFTA), CCT La Plata Consejo Nacional de Investigaciones Científicas y Técnicas (CONICET), Universidad Nacional de La Plata (UNLP), Diagonal 113 y 64 S/N, La Plata 1900, Argentina; fiorelag@inifta.unlp.edu.ar (F.G.); edprieto@quimica.unlp.edu.ar (E.P.); 3Instituto de Ciencias de la Salud, Universidad Nacional Arturo Jauretche (ICS-UNAJ), Avenida Calchaqui, Florencio Varela 6200, Argentina; 4Laboratorio de Bioproducción, Plataforma de Bioinsumos, Instituto Nacional de Investigación Agropecuaria, Estación Experimental Wilson Ferreira Aldunate, Ruta 48 Km 10, Canelones 90000, Uruguay; hoberti@inia.org.uy (H.O.); eabreo@inia.org.uy (E.A.); 5Departamento de Biotecnología, Universidad ORT Uruguay, Montevideo 10129, Uruguay

**Keywords:** entomopathogenic fungi, insect gut, oral infection, coleoptera, EGFP, alginate, AFM

## Abstract

The ability of entomopathogenic fungi, such as *Beauveria bassiana*, to infect insects by penetrating their cuticle is well documented. However, some insects have evolved mechanisms to combat fungal infections. The red flour beetle (*Tribolium castaneum*), a major pest causing significant economic losses in stored product environments globally, embeds antifungal compounds within its cuticle as a protective barrier. Previous reports have addressed the contributions of non-cuticular infection routes, noting an increase in mortality in beetles fed with conidia. In this study, we further explore the progression and dynamics of oral exposure in the gut of *T. castaneum* after feeding with an encapsulated *B. bassiana* conidia formulation. First, we characterized the formulation surface using atomic force microscopy, observing no significant topological differences between capsules containing and not containing conidia. Confocal microscopy confirmed uniform conidia distribution within the hydrogel matrix. Then, larvae and adult insects fed with the conidia-encapsulated formulation exhibited *B. bassiana* distributed throughout the alimentary canal, with a higher presence of conidia before the pyloric chamber. More conidia were found in the larval midgut and hindgut compared to adults, but no germinated conidia were observed in the epithelium. These results suggest that the presence of conidia obstructs the gut, impairing the insect’s ability to ingest, process, and absorb nutrients. This disruption may weaken the host, increasing its susceptibility to infections and, ultimately, leading to death. By providing the first direct observation of fungal conidia within the alimentary canal of *T. castaneum*, this study highlights a novel aspect of fungal–host interaction and opens new avenues for advancing fungal-based pest control strategies by exploiting stage-specific vulnerabilities.

## 1. Introduction

Entomopathogenic fungi such as *Beauveria bassiana* and *Metarhizium* spp. infect insects by penetrating the cuticle, a route not used by other microbial pathogens like protozoa, bacteria, or viruses [1]. During infection, these fungi release enzymes and toxins that compromise the host’s immune system [2,3]. Several studies have identified specific fungal genes involved in these processes, although the molecular mechanisms involved remain under investigation [4,5,6,7,8,9]. Also, entomopathogenic fungi can interact with the insect gut microbiome, affecting the insect’s immunity and gut permeability [10,11]. These interactions are pivotal in overcoming host defenses and facilitating infection. Insects exhibit various defensive behaviors and physiological adaptations against fungal infections. Some insects alter their feeding or grooming behavior to avoid spores, while others strengthen their immune defenses [9]. One example of evolved immunity strategies is the production of antifungal compounds, which are incorporated into their cuticles, making them less susceptible to fungal infections [12,13,14,15,16,17,18,19]. Using this strategy, adults of the red flour beetle *Tribolium castaneum* exhibit tolerance to *B. bassiana* infection via cuticular penetration [18,19,20,21].

Among non-canonical infection routes, ingestion of fungal propagules was first proposed more than a century ago [22], and there is more recent evidence demonstrating that entomopathogenic fungi can achieve infection through routes other than cuticular penetration [23]. In various insect species, the ultimate destination of ingested conidia varies, as does the potential for internal adhesion and germination [22]. Research on oral infection in Coleoptera suggests that fungal invasion could start through the mouthparts of different beetle species, often with hyphal growth, but there are no reported cases of germination inside the digestive tracts [22]. Thus, the fate of conidia following ingestion in *T. castaneum* remains largely unknown. Recently, we investigated dual routes of infection—combining cuticular penetration with the ingestion of conidia incorporated into the insect’s diet. This approach highlighted the significant contribution of alternative infection pathways, leading to increased mortality and activation of immunity-related gene clusters [3,24]. *T. castaneum* feeding habits do not differ drastically between larvae and adults; however, there are key anatomical differences in the alimentary canal related to cell renewal and food storage [25]. Aspects of the beetle’s feeding behavior and alimentary canal anatomy may facilitate the establishment of fungal conidia ingested with the diet, either by enabling infection or by sufficiently disrupting nutrient absorption to adversely affect the insect’s life cycle, ultimately resulting in reduced fitness [26].

Fungal formulations incorporating gelling agents, such as sodium alginate, are widely used to protect the active agent from environmental stressors, thereby extending the release and efficacy of fungal conidia. This characteristic presents a valuable opportunity to ensure that conidia come into contact only through ingestion after these formulations are offered in the beetle diet.

In this study, we propose the use of a conidial hydrogel formulation to characterize the distribution and behavior of conidia within larvae and adult *T. castaneum*. Feeding insects with conidia embedded in the diet enabled tracking of conidial movement through the alimentary canal using GFP-tagged *B. bassiana*. Fungal spores were detected throughout the alimentary canal in both larvae and adult insects, with a uniform distribution but a higher concentration in the hindgut (ileum) region of both larvae and adults. These findings indicate that ingested conidia are unable to germinate within the alimentary canal but exert a measurable impact on the insects’ fitness, increasing susceptibility to fungal infection and eventual mortality.

## 2. Materials and Methods

### 2.1. Fungi

The entomopathogenic fungus *B. bassiana* strain ARSEF 2860 (ARSEF is WDCM collection number 112) was selected for this study. The strain was provided by R. Humber’s Lab (Plant Soil and Nutrition Laboratory, USDA-ARS, Ithaca, NY, USA). To obtain conidia, the fungus was maintained and routinely grown at 26 °C for 15 days in Potato Dextrose Agar (PDA; Britania, Buenos Aires, Argentina) medium with 1% ampicillin plates. Sporulating plates were harvested, and fungal propagules were processed for downstream assays. 

#### 2.1.1. Enhanced Green Fluorescent Protein (EGFP) *Beauveria bassiana*

The binary vector pCGEN-GFP [27] containing the enhanced green fluorescent protein (EGFP) [28] and the geneticin resistance gene (neo) was used for *Agrobacterium tumefaciens*-mediated transformation of EGFP in *B. bassiana* ARSEF 2860, as described in Oberti et al. [29]. The best transformant strain was selected by fluorescence intensity quantitation in the 488 channel under a Carl Zeiss Observer LSM 800 Confocal Microscope (Oberkochen, Germany) with a 40X NA 0.75 Objective. The stability of the selected transformants was evaluated through serial passaging in Czapek-Dox medium (Merk, Darmstadt, Germany) with G-418 (neoR) (700 μg/mL) (Sigma-Aldrich, St. Louis, MO, USA) for ten consecutive generations to ensure their ability to maintain the genetic modification. Following this, the transformants were transferred to PDA medium without G418 to assess phenotypic stability in comparison to the wild-type strains. The best transformant was selected based on the results of virulence bioassays, which compared the GFP strain to the wild-type strain using model insects, as well as the quantification of fluorescence intensity in the 488 channel under a Carl Zeiss Observer LSM 800 Confocal Microscope with a 40× NA 0.75 Objective.

#### 2.1.2. Conidia Powder Preparation

Harvested conidia were dried for 7 days at 26 °C in a 35 mm Petri dish with sterile filter paper, which was placed inside a 90 mm Petri dish filled with silica gel, and allowed to dry in an incubator. Once dried, the propagule was ground with a mortar and pestle into a uniform powder and stored at 4 °C. The number of conidia per gram of conidia powder was determined in a Neubauer chamber after resuspending 2 mg of conidia in a sterile solution of 0.05% Tween 80. Viability was verified using the quantified conidia suspension.

### 2.2. Insects

Larvae and adults of *T. castaneum* from colonies maintained at the Instituto de Investigaciones Bioquímicas de La Plata (INIBIOLP) were used. Insects were reared on white wheat flour, with 5% non-fat dried milk, 5% brewer’s yeast, and 5% wheat germ, under a 12L:12D photoperiod at 27 ± 2 °C and 70 ± 5% relative humidity. All larvae were selected as last instar larvae (4.5 ± 0.5 mm length), unsexed insects. All adults used in the assays were 2-week-old, unsexed insects. 

### 2.3. Conidia-Encapsulated Formulation

Hydrogel capsules were prepared using a mixture of 1 mL of 1% sodium alginate, 0.1 g of wheat flour, and conidia powder, with the concentration adjusted to 1 × 10^9^ conidia/mL of mixture. To form the capsules, a 10 mL medical-grade syringe with a 21 G × 1″ hypodermic needle was used. The mixture was extruded dropwise over a 0.2 M calcium chloride solution in a beaker, which was stirred continuously to ensure uninterrupted capsule formation. The mixture droplets instantaneously formed capsules upon contact with the calcium chloride solution and were left to consolidate under agitation for 40 min. The capsules were then rinsed three times in sterile distilled water and placed on sterile filter paper to absorb excess water. Immediately afterward, the capsules were dried for 3 days as described before for conidia. Controls were prepared similarly, but with sterile distilled water replacing the conidia powder. Yield was calculated as the number of capsules obtained per ml of sodium alginate used. The size of the encapsulated formulation was evaluated using a Zeiss Primo Star optical microscope (Jena, Germany) (40×, NA 0.75). A total of 100 capsules were used, and three independent experiments were conducted.

### 2.4. Capsules Characterization

#### 2.4.1. Surface Topography

For formulation topography characterization, atomic force microscopy (AFM) images of freshly prepared capsules were obtained in air using a MultiMode Scanning Probe Microscope (Veeco, Plainview, NY, USA) equipped with a Nanoscope V controller (Veeco). All measurements were performed in Tapping^®^ mode. Images were captured at 1 Hz with silicon tips (RTESP, 215–254 kHz and 20–80 N/m). Roughness data (Ra) were obtained from four separate images taken from different regions of each substrate using Nanoscope V Software 9.7. Ra was calculated as the arithmetic average of the absolute values of the surface height deviations from the mean plane.

#### 2.4.2. Conidial Distribution

Confocal microscopy images of capsules were used to assess the distribution of conidia inside the hydrogel matrix. The capsules were made with *B. bassiana* ARSEF 2860 EGFP conidia, hydrated, sectioned using a stereoscopic microscope (Zeiss, Jena, Germany, Stemi 305, 0.8x–5x) with a scalpel blade at 0.5 mm, and mounted in 100 µL of sterile physiological solution on a microscope slide. A coverslip was placed on top and sealed with nail polish. Controls with no conidia and non-EGFP conidia were treated similarly. Confocal microscopy images were acquired using a Carl Zeiss Observer LSM 800 Confocal Microscope (Objectives: 10X NA 0.3, 40X NA 0.75, 63X oil immersion NA 1.4). A total of five images per condition were taken for representative and quantitative purposes.

#### 2.4.3. Capsules Stress Tolerance

Thermal tolerance: Individual capsules were placed in 100 µL of 0.05% Tween 80 and incubated for 1, 3, and 5 h at 45 °C in a heat block. Controls were prepared using capsules without conidia. Afterward, capsules were homogenized using a pellet pestle, and 100 µL of the homogenate was plated onto a Sabouraud Dextrose Yeast Extract Agar (SDYA) plate. Samples were incubated for 10 days at 26 °C. Conidia viability was assessed by collecting fungal sporulation from each plate in 5 mL of 0.01% Tween 80, diluting, and quantifying the conidia in a Neubauer chamber. The experiment was repeated three times.

Oxidative stress tolerance: Individual capsules were placed in 1 mL of 12 mM H_2_O_2_ solution and kept at 4 °C for 72 h. As controls, the same amount of conidia powder and capsules without conidia in 1 mL of 12 mM H_2_O_2_ were used. Each sample was centrifuged at 13,000 rpm for 1 min, and 900 µL of the supernatant was discarded. The remaining solution was homogenized. Next, 100 µL of each sample was plated onto SDYA with ampicillin (50 µg/mL) and chloramphenicol (34 µg/mL). The samples were incubated for 10 days at 26 °C. Conidia viability was assessed by collecting fungal sporulation from each plate in 5 mL of 0.05% Tween 80, diluting, and quantifying the conidia in a Neubauer chamber. The experiment was repeated three times.

#### 2.4.4. Capsules Pathogenicity Assay

Three groups of 10 *T. castaneum* individuals (either adults or larvae) were placed in sterile plates containing capsules, accounting for a total of 1 × 10^9^ conidia. Control plates were prepared similarly, using capsules without conidia. The individuals were maintained at 26 °C for 15 days. Mortality was checked daily, and all dead adults and larvae were removed after each count. Subsequently, the dead insects were washed in 10% bleach for 30 s, then in 70% ethanol for 30 s, rinsed in sterile distilled water, allowed to dry on sterile filter paper, and placed in individual humid chambers at 26 °C to confirm fungal infection. The experiment was repeated three times, and mortality data were corrected for control mortality using Abbott’s formula [30].

#### 2.4.5. Conidia Ingestion Assay and Dissection

Capsules used for feeding were made with *B. bassiana* ARSEF 2860 EGFP, as described in Section 2.4. *T. castaneum* adults and larvae were placed in groups of 10 insects in sterile plates containing capsules, accounting for 1 × 10^9^ conidia/g. Control plates were prepared similarly, using capsules without conidia. The individuals were maintained at 26 °C for 72 h. Afterward, the insects were dissected in sterile physiological solution on a clean microscope slide. The alimentary canals were extracted and placed in 50 µL of physiological solution.

#### 2.4.6. Cuticular Infection Assay and Dissection

Wheat flour and fungal powder (1 × 10^9^ conidia), without encapsulation, were used to feed the insects under the conditions described in Section 2.4.5. Controls were prepared by replacing the conidia with wheat flour. The same experimental procedures were followed for dissecting the alimentary canal for observation.

### 2.5. Colony-Forming Unit Counts

The alimentary canals were placed in physiological solution, vigorously vortexed for 1 min, diluted to a final volume of 1 mL, and plated onto PDA leaving the alimentary canal out. All samples were incubated at 26 °C for 7 days, and colony-forming unit (CFU) counts were recorded. Controls were treated in the same manner. The experiments were repeated three times. All CFUs were examined for fluorescence using confocal microscopy.

### 2.6. Confocal Microscopy

Dissected alimentary canals or insect heads were placed in 20 µL of physiological solution on a clean microscope slide, sealed with a coverslip and nail polish for observation. Imaging was performed using a Carl Zeiss Observer LSM 800 Confocal Microscope, equipped with 10X, 40X plan-neofluar air objectives and a 63X apochromat oil objective (NA 0.3, 0.75, and 1.4, respectively), controlled by ZEN software 3.11. Three images were captured for each region and condition. Alimentary canal sections were imaged with the different objectives and digital zoom at 488 nm and 561 nm channels. Laser power settings were 2.51% for the 561 nm channel and 2% for the 488 nm channel (pinhole 459 nm).

### 2.7. Data Analysis

All statistical analyses and graphs were performed using GraphPad Prism version 10.0.0 for Windows (GraphPad Software Inc., San Diego, CA, USA; www.graphpad.com). Image analysis was conducted using Fiji (ImageJ 2.14.0; http://imagej.net/ij (accessed on 15 December 2024)) [31]. Intensity quantification was performed by setting five lines (Regions of Interest, ROI) across the image in the 488 nm and 561 nm channels, and the intensity distribution was graphed against distance. Conidia counts were performed manually using the rectangle tool to define four areas per image from three independent images. Counts were made using the Multi-point tool and were normalized by area.

## 3. Results

### 3.1. Morphological Characterization of Capsules and Conidia Distribution Within the Matrix

The morphological characterization of alginate capsule formulations was performed using optical microscopy. While initially spherical, the capsules flattened into a disc shape upon drying (Appendix A). The hydrated and dried capsules, both control and conidia-containing, showed no significant differences in size (Appendix A) or yield (97 ± 13 capsules/mL).

Surface topology analysis using AFM revealed no significant differences in any of the analyzed parameters between control and conidia-containing capsules (Figure 1A,B and Appendix A), indicating that conidia do not alter the surface properties relevant for beetle feeding, at least within the resolution limits of the imaging method. It also confirms that conidia are trapped in the hydrogel matrix, reducing the risk of cuticular infection.

Longitudinal sections of the hydrogel capsules confirmed uniform conidia distribution throughout the matrix. Both control (Figure 1C–F) and *B. bassiana*-containing capsules (Appendix A) exhibited minimal autofluorescence. Fluorescence was much more pronounced in EGFP-tagged *B. bassiana* (Figure 1G–J). The EGFP-tagged conidia displayed peaks in the 488 nm (green) channel, with additional autofluorescence detected at 561 nm (red), consistent with fungal autofluorescence (Appendix A).

Thermal and oxidative stress assays further demonstrated that the alginate formulation significantly outperformed the dried conidia powder under all tested conditions, highlighting its enhanced protective capabilities (Appendix A).

Potential cuticular infection during feeding was also tested by imaging the heads of *T. castaneum* after they were fed hydrogel-flour conidia capsules and flour-(naked) conidia powder. There was no evidence of conidia adhesion in the head cuticle in controls (Figure 2A–D) nor on the cuticle of beetles fed with hydrogel conidia capsules (Figure 2E–H), in contrast to those fed with naked-conidia powder (Figure 2I–L), where a significant green signal was shown (Figure 2I–L, red arrow), allowing the green signal to be positively identified as conidia.

### 3.2. Capsules Virulence and Conidia Presence in the Alimentary Canal of T. castaneum

To evaluate the availability and virulence of the alginate hydrogel capsules, cumulative mortality assays were conducted. Larval and adult *T. castaneum* individuals were fed either conidia-containing capsules or control capsules. Larvae began to exhibit mortality as early as day 2 of feeding, whereas adult mortality was observed from day 5 onwards (Figure 3A). Cumulative mortality was also measured for the dried conidia powder used to prepare the capsules (Figure 3B). For larvae, mortality rates were comparable between the encapsulated and powder formulations, demonstrating that the hydrogel capsules effectively delivered *B. bassiana* conidia to this life stage. However, in adults, dried conidial powder induced twice the cumulative mortality compared to the capsules, suggesting that conidia powder provides greater efficacy in adult *T. castaneum*. Consistent with previous findings, adult beetles showed greater tolerance to *B. bassiana* infection, with cumulative mortality rates lower than those of larvae across both delivery methods. This disparity likely reflects differences in susceptibility between life stages, such as variations in immune defenses or cuticle composition. Despite the relatively lower mortality in adults, humid chamber assays performed on cadavers revealed fungal growth in all dead insects, indicating that *B. bassiana* confined to the hydrogel capsule successfully infected and killed both larvae and adults.

To confirm the ingestion and localization of conidia within the alimentary canal, dissections were performed, and the alimentary canal contents were plated to evaluate colony-forming units (CFUs). EGFP-tagged colonies were recovered in both larvae and adults, demonstrating effective ingestion (Appendix A). The recovery of EGFP-tagged *B. bassiana* from the alimentary canal confirms that the alginate hydrogel capsules served as an effective vehicle for conidial delivery.

### 3.3. Conidia Visualization in T. castaneum Alimentary Canal

The insect alimentary canal is divided into three main regions: the foregut, midgut, and hindgut, each lined by a single layer of epithelial cells (Figure 4A,B). To better understand the fate of conidia following ingestion, the alimentary canal of *T. castaneum* was imaged using confocal microscopy.

Autofluorescence in the gut tissue was initially characterized, revealing strong signals, predominantly in the 561 nm (red) channel (Appendix A). Among the gut structures, the pyloric chamber cells (PyC) were distinctly identifiable in both larval and adult *T. castaneum* and were selected as reference points for imaging.

To examine the distribution of EGFP-tagged conidia within the alimentary canal of *T. castaneum* larvae, confocal images were acquired after feeding with conidia-containing hydrogel capsules. After identification of the PyC, imaging was performed on the midgut (MG) and ileum (IL) regions (blue arrows, Figure 4B). EGFP-tagged conidia were observed in large quantities surrounding the PyC, appearing as individual spherical cells or cell clusters (red arrows, Figure 4C–E). Fluorescence intensity quantification revealed distinct peaks in the green channel, corresponding to the presence of conidia, while the red channel peaks indicated autofluorescence of the PyC (Figure 4F). Conidia were clearly visible on the green channel (Figure 4E) but absent in the red channel (Figure 4D) in regions highlighted by red arrows. In control larvae, minimal to no signal was detected in the green channel, although the PyC autofluorescence was evident in the red channel (Figure 4G–J).

To further assess the localization and morphology of conidia, higher magnification images of the MG and IL regions were obtained from both control and conidia-fed larvae (Figure 5). As expected, MG control samples showed no detectable conidia in the MG (Figure 5A–C), as is reflected in the absence of signal in the green channel (Figure 5D). In contrast, larvae fed with EGFP-tagged conidia exhibited abundant fungal presence in the MG, with individual conidia clearly visible (Figure 5E–G, red arrow). Quantitative fluorescence analysis showed peaks in the green channel corresponding to the observed conidia (Figure 5H). For the IL region control samples, no conidia can be detected (Figure 5I–K) and a minimal fluorescent signal was detected in the green channel (Figure 5L). Similarly to the observations for the MG region, in the IL region of larvae fed with EGFP-tagged conidia, individual spherical conidia were detected (Figure 5M–P), with an increase in the green signal as well (Figure 5P). Conidia counts in the analyzed alimentary canal regions revealed significant differences between the MG and hindgut (HG) for both larvae and adults (Appendix A). In larvae, conidia tended to accumulate more in the HG compared to the MG, while in adults, a higher number of conidia were found in the MG compared to the HG (Appendix A). A closer inspection of the alimentary canal lumen revealed no evidence of germinating conidia in either the MG or IL regions, suggesting that the conidia remained viable but ungerminated within the gut environment.

The alimentary canal of adult *T. castaneum* exhibits greater anatomical and secretory complexity compared to larvae, as previously described [24]. Notable structural features include a more melanized PyC (Figure 6A–C), the MG lined with finger-like projections known as regenerative crypts—clusters of regenerative cells (Figure 6B)—and the distinctly organized ileum (IL) and colon (CL) regions (Figure 6C). These areas were used as reference points for fluorescence imaging in adults.

In control adults, the PyC was the only distinct structure observed (Figure 6D–F), emitting low autofluorescence signals primarily detected in the red channel (Figure 6G). In contrast, adults fed EGFP-tagged conidia capsules displayed a significant accumulation of conidia within the MG. Groups of EGFP-tagged conidia were clearly visible, often mixed with the formulation matrix (Figure 6H–J). A specific conidia cluster, marked by a red arrow, is shown in the merged image (Figure 6H), absent in the red channel (Figure 6I), but prominent in the green channel (Figure 6J). Quantitative fluorescence analysis confirmed intensity peaks exclusively in the green channel (Figure 6K). 

Similar observations were made in the ileum (IL) region (Figure 6L–N). Conidia clusters, marked by red arrows in panels L-N of Figure 6, were detected only in the green channel, with corresponding intensity quantification shown in Figure 6O. In Figure 7, higher magnification images of the MG and HG regions were analyzed. Controls for the MG (Figure 7A–D) and HG (Figure 7I–L) showed only the characteristic low autofluorescence previously observed in the insect. In the adult MG, several conidia clusters were visible (red arrows, Figure 7E–H). Conidia presence is also reflected in intensity quantification, with peaks in the green channel (Figure 7H). Similarly, in the HG, clusters and individual conidia were clearly identifiable (red arrows, Figure 7M–P), with corresponding green channel peaks observed (Figure 7P). Consistent with findings in larvae, the conidia count in adults did not show significant differences between the MG and IL regions (Appendix A). These results suggest that the distribution and accessibility of conidia are comparable across these alimentary canal regions in both developmental stages.

When beetles were exposed to a mixture of wheat flour and conidia powder without the gelling agent, a very different scenario emerged. As shown in Figure 8, there were no conidia in controls (Figure 8A–D). In contrast, conidia were only evident in the larval midgut (Figure 8E–H). The distribution of conidia appeared to differ from that observed when conidia were encapsulated (Figure 4), with no germination observed. For adults, no conidia were detected in the alimentary canal. Neither fungal presence nor significant differences between control (Figure 8I–L) and treated samples (Figure 8M–P) were observed. We can conclude that this delivery method is ineffective for oral infection in adults but may be relevant for larvae.

## 4. Discussion

Entomopathogenic fungi are well-known for infecting insects through cuticular adhesion and penetration, while oral infection via spore ingestion remains comparatively unexplored. This route becomes particularly relevant when fungal conidia are ingested along with food sources, such as grains and their derived products, by insect pests like the red flour beetle *T. castaneum*, which poses a significant threat to agricultural productivity, especially in developing countries. Previous studies have documented *B. bassiana’s* potential to infect through oral ingestion, as its spores can bypass the alimentary canal defenses to reach internal tissues, as shown in other insect species [3,32,33,34,35]. This study aimed to investigate the potential of *B. bassiana* infection via the oral route in *T. castaneum*, focusing on its interaction within the insect’s alimentary canal.

While oral fungal infections are often associated with spore germination and tissue invasion, recent studies suggest alternative mechanisms of mortality. For instance, in *Metarhizium anisopliae*, mortality in mosquito larvae has been attributed to protease-induced apoptosis rather than direct cuticular or gut invasion, highlighting the role of fungal enzymes in disrupting host defenses [36]. Similarly, *B. bassiana* may induce mortality through proteolytic enzymes, which degrade structural components of the insect gut and impair cellular immunity. This aligns with our observation that while conidia germination was not detected in the alimentary canal, mortality still occurred, potentially due to enzymatic activity or secondary metabolites disrupting nutrient absorption.

Hydrogel encapsulation, which is used to protect and extend the viability of fungal conidia, provided significant advantages for targeted pest control [37]. This formulation controlled conidial release, minimized premature germination, and provided a monitored environment for studying conidial distribution. The encapsulation process successfully ensured a uniform distribution of conidia, as confirmed by atomic force and confocal microscopies, and did not hinder their viability or infection potential. Encapsulated conidia were specifically accessible through ingestion, minimizing unintended exposure and preventing conidial adhesion to the beetle’s cuticle. However, the reduced efficacy in adults suggests the need for further optimization of the capsule formulation to enhance its virulence in more resistant life stages. Notably, no fungal deposition was observed on the heads of beetles exposed to hydrogel capsules, contrasting with the conidia-flour mixture treatment, further highlighting the effectiveness of encapsulation in preventing cuticular adhesion. Although conidia ingestion via hydrogel capsules led to mortality in *T. castaneum*, the underlying mechanisms driving fungal infection following ingestion remain unclear. The current study traced the path of fluorescent *B. bassiana* conidia after ingestion and hypothesized whether these fungi might germinate within the gut. While previous studies on other insect species like *Bombyx mori* [38] documented hyphal growth from the gut, similar observations have not been reported in Coleoptera species such as *T. castaneum* [23,39].

In this study, conidia were more concentrated in the hindgut of larvae, with fewer conidia recovered from the midgut. The hindgut is a narrower section of the alimentary canal, which alone can account for a higher count of conidia per area. For adult beetles, the analysis was somewhat more complex. While the number of conidia counted in the midgut was similar to that in larvae, a reduction in the hindgut compared to both the larval hindgut and the adult midgut was observed. The observed reduction in hindgut conidia compared to larvae may reflect faster food passage and more efficient gut defenses, as seen in other insect–pathogen interactions [36]. This suggests that gut anatomy and feeding behavior may influence fungal uptake and distribution. The more developed musculature and faster food passage in adults likely contributed to a reduced fungal retention time in the gut, thus limiting fungal adhesion opportunities. Additionally, the adult midgut’s regenerative crypts, which house intestinal stem cells (ISCs) that rapidly renew the gut epithelium, may complicate conidia adhesion and colonization. ISCs are known to play a role in immunity, yet their involvement in the response to entomopathogenic fungi in Coleoptera remains poorly understood [25]. Recent RNA-seq analysis of *B. bassiana*-infected beetles revealed Toll receptors and members of the JNK pathway but no evidence of antimicrobial peptides (AMPs), suggesting that the rapid passage of conidia might limit immune responses [3].

Another key point that has to be considered is the presence of secondary metabolites secreted by *B. bassiana*. After reaching the nutrient-rich inside of the host, the fungal mycelium transitions into a specialized yeast-like cell phenotype known as hyphal bodies, which secrete a diverse array of toxic secondary metabolites [2,40]. These compounds act as immunosuppressants and infection facilitators [41], with distinct roles depending on tissue (live or dead) and infection state [42,43,44]. For example, destruxins produced by *Metarhizium anisopliae* induce paralysis, disrupt antioxidant defenses, suppress the humoral immune system, and exhibit phagodepressant activity, contributing to host starvation [7,45]. Similarly, *B. bassiana* secondary metabolites such as beauvericins and beauverolides impair phagocytosis and immune cell movement, further weakening the host immune response [46,47,48,49]. The high abundace of conidia observed in the alimentary canal of both larvae and adults in this study, along with the associated mortality rates, may also be linked to the secretion of these toxic metabolites by fungal spores. Further research is required to identify the specific secondary metabolites involved in oral infection and to clarify their roles in the pathogenesis of *B. bassiana*.

The insect gut is protected by the peritrophic membrane, a chitin-based structure reinforced with proteins and glycoproteins [25], which may physically hinder fungal adhesion and limit contact with gut tissues. Beyond serving as an “exclusion sieve”, the glycoproteins in the peritrophic membrane provide binding sites for various pathogens, restricting their access to the midgut epithelium [50]. Fungal entomopathogens such as *B. bassiana* have evolved mechanisms to breach this barrier, producing chitinases and enterotoxins, which bind to specific sites on the peritrophic membrane in Diptera and Lepidoptera [50,51]. However, this binding is influenced by the developmental stage and insect species, as the peritrophic membrane can sequester protein toxin receptors and limit the penetration of dissolved endotoxin aggregates [52,53]. Additionally, membrane-associated proteases can cleave protein toxins, further contributing to resistance [53,54].

Furthermore, digestive enzymes like proteases, lipases, and chitinases counteract fungal infection by degrading fungal cell walls, particularly the chitin matrix [55]. This could potentially explain the reduction in conidia number observed in the adult hindgut. In adult *T. castaneum*, genes involved in chitin metabolism were significantly affected when *B. bassiana* conidia was added to the insect diet [3]. This complex dynamic interaction between the fungal pathogen trying to breach the peritrophic membrane and the insect’s defensive response likely contributes to the developmental stage differences observed in this study.

The gut environment also influences fungal pathogenesis. The alkaline environment of the insect gut may also exacerbate fungal degradation, as optimal fungal germination conditions typically fall within a pH range of 6–7, while the insect gut’s pH may hinder fungal germination and pathogenesis [56]. Moreover, the microbiota present in the insect gut can play a critical role in immune response and resistance to entomopathogenic fungi [57]. For example, *B. bassiana* infection accelerates mortality in individuals with specific gut microbiota, while bacteria such as *Wolbachia* and antifungal-producing *Burkholderia* confer resistance to this fungus [58,59]. The role of microbiota in oral fungal infection also remains unclear in *T. castaneum*, as prior studies did not detect AMPs after oral infection, although activation of the Toll pathway was evident [3]. Despite the presence of specific insect defenses, *B. bassiana* has evolved strategies to overcome gut defenses, such as producing inhibitors of peptidases [3] and serine proteases [56] to neutralize digestive enzymes in the midgut. This resistance might explain the difference in conidial recovery between larvae and adults. As discussed below, the gut environment and the insect’s immune system may influence fungal adhesion and colonization.

The differences observed when encapsulated conidia were fed to beetles, in contrast to naked-conidia powder, were more striking in adults. Although conidia adhered to the head of adult beetles were visualized, no conidia were detected in the alimentary canal. This could be due to adult *T. castaneum* repelling most of the conidia by preventing adhesion to the cuticle [18], or at least to an extent where it cannot be detected in the gut. The gut conditions may also influence the naked conidia, making them more susceptible to the intestinal environment, such as harsh pH conditions or the presence of digestive enzymes. This is an interesting question that requires further research to be answered.

The observed differences in conidial distribution and infection outcomes between larvae and adults of *T. castaneum* likely contribute to varying mortality rates in each developmental stage following fungal exposure. While both larvae and adults were susceptible to *B. bassiana* infection via oral ingestion, mortality rates were higher in larvae compared to adults. This could be attributed to several factors, including differences in immune response, gut anatomy, and digestive processes. The immune response at different insect stages may play a key role, as mortality rates from naked conidia via cuticular infection are also higher in larvae compared to adults [17,18,19]. The larvae immune defenses, including cellular responses, may be less effective at neutralizing fungal invaders, enabling a higher rate of infection and mortality. On the other hand, the alimentary canal of larvae is less developed, and their slower feeding behavior may allow for greater retention of conidia within the gut, increasing the likelihood of fungal colonization. Conversely, adult beetles exhibit more complex and efficient immune responses. As mentioned earlier, the presence of regenerative crypts and faster food passage could reduce the retention time of conidia in the gut, limiting fungal adhesion and colonization. The more developed immune system of adults may also play a crucial role in limiting fungal spread. For example, the activation of the Toll pathway, as observed in previous studies [3,22], could help adults mount a more rapid immune response to prevent fungal proliferation. Furthermore, the peritrophic membrane, which is more developed in adults, may serve as an additional physical barrier to fungal attachment and penetration, further protecting the beetles from infection. However, despite these defenses, adults still experience some level of mortality when exposed to *B. bassiana*. This suggests that the physiological mechanisms at play—such as immune responses, gut structure, and digestion—are not entirely sufficient to prevent fungal infection, particularly when the conidia are ingested via formulations like hydrogel capsules.

Overall, these findings highlight that fungal virulence and the effectiveness of fungal-based pest control strategies may be highly stage-dependent. The increased tolerance of adults may lead to lower mortality rates compared to larvae; however, this does not diminish the potential for significant pest control effects, particularly in larval populations. Further research into the immune dynamics, gut physiology, and microbial interactions across different developmental stages of *T. castaneum* will provide valuable insights for optimizing fungal formulations and enhancing pest management strategies.

These findings suggest that although conidia ingested through hydrogel capsules do not germinate within the alimentary canal of either larvae or adults, they still disrupt the insect’s ability to process and absorb nutrients. This blockage or competition may contribute to mortality by compromising the insect’s overall health. Future research should focus on elucidating the physiological mechanisms behind stage-dependent differences in fungal infections and optimizing fungal formulations for more effective pest control.

In conclusion, this study provides the first direct observation of conidia within the alimentary canal of *T. castaneum*. While conidia germination was not detected in the gut, their presence seems to contribute to mortality by impairing nutrient absorption. These results highlight the potential of encapsulated formulations in advancing fungal-based pest control strategies, opening a promising avenue for infection via fungal ingestion.

## Figures and Tables

**Figure 1 jof-11-00101-f001:**
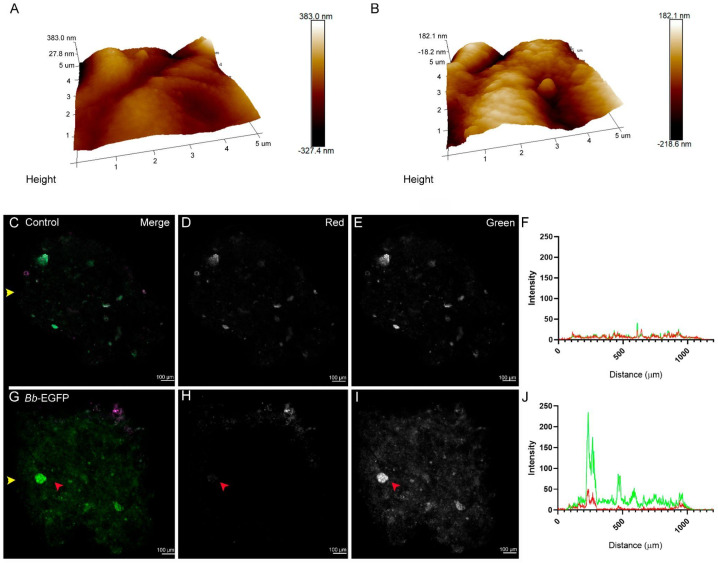
Microscopic characterization of *B. bassiana* alginate capsules. Control (**A**) and conidia-containing (**B**) capsule topologies were characterized using atomic force microscopy (AFM) over a 5 µm area. No significant differences were found. Confocal images of control (**C**–**E**) and conidia-containing (**G**–**I**) capsules are shown. Intensity quantification confirms the absence of green signal peaks (**F**), while (**J**) shows significantly higher green peaks in conidia-containing capsules. In (**F**–**J**), the green and red lines represent the green and red channel signals, respectively. Channels are indicated in the top-right corner of the top panels. Red arrows indicate conidia, and yellow arrows indicate the horizontal line where signal quantification was measured. Scale bar for confocal microscopy images represents 100 µm. Sample treatment is indicated at the top left corner.

**Figure 2 jof-11-00101-f002:**
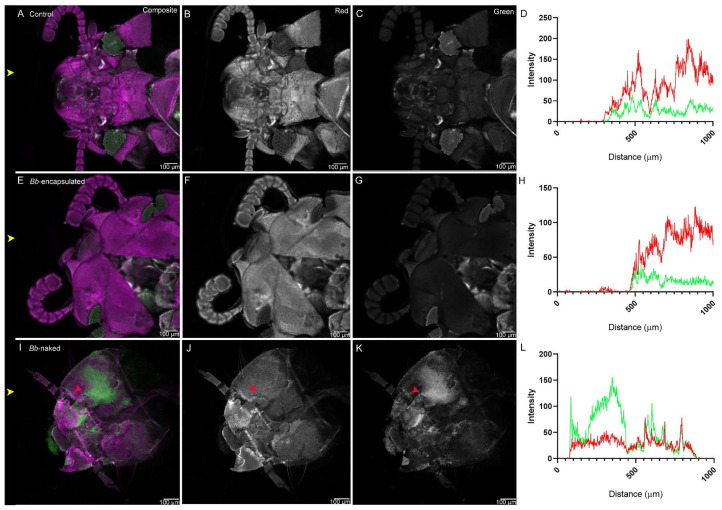
Adult *T. castaneum* heads after feeding. Autofluorescence in both the red and the green channels in control samples that were not in contact with fungi (**A**–**D**). Adults fed encapsulated conidia showed no green signal on the head cuticle (**E**–**H**), indicating that no conidia were detected. For beetles fed naked-conidia powder (**I**–**L**), an increase in the green signal was detected on the cuticle, indicating conidia adhesion (red arrow). Fluorescent signal quantification is represented in panels (**D**,**H**,**L**), green lines represent signals in the green channel, while red lines represent signals in the red channel. The channels are indicated in the top-right corner of the top panels. Yellow arrows indicate the horizontal line where signal quantification was measured. Scale bar for confocal microscopy images represents 100 µm. Sample treatment is indicated at the top left corner.

**Figure 3 jof-11-00101-f003:**
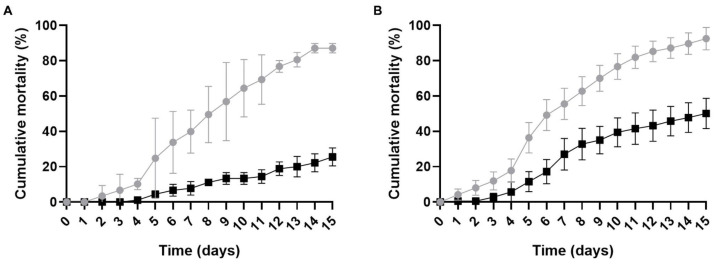
Cumulative mortality of larvae and adults of *T. castaneum* fed with *B. bassiana.* (**A**) Cumulative mortality for capsules. (**B**) Cumulative mortality for conidia powder. Larvae are represented in grey, and adults in black. Mortality rates were corrected using Abbott’s formula.

**Figure 4 jof-11-00101-f004:**
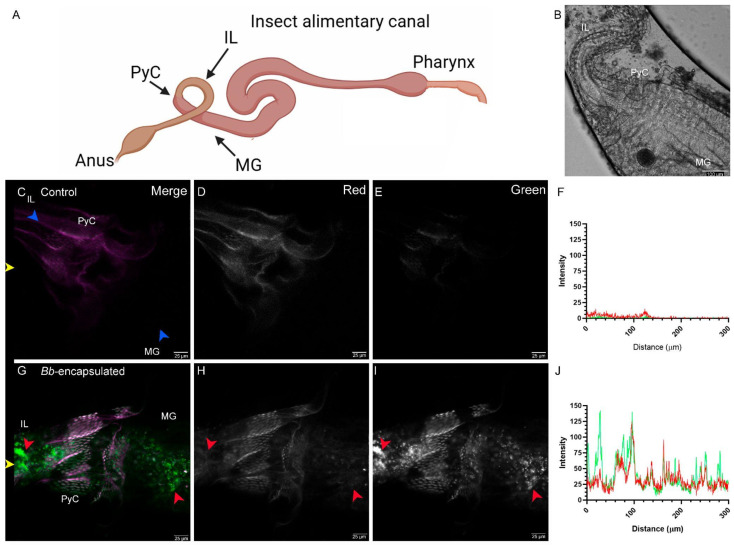
Presence of conidia in the alimentary canal of larval *T. castaneum* fed alginate capsules containing *B. bassiana* conidia. (**A**) Diagram of the insect alimentary canal structure with references to imaging locations. (**B**) Bright-field image of *T. castaneum* larvae showing pyloric chamber cells (PyC), midgut (MG), and ileum (IL) sections. (**C**–**F**) Control larvae show signal only in the red channel, while in (**G**–**J**), larvae fed conidia exhibit a clear green signal throughout the different regions of the alimentary canal. In (**F**,**J**), fluorescent signal quantification graphs are shown, and green and red lines indicate signals in the green and red channels, respectively. Channel information is provided in the top-right of the top panels. Red arrows indicate conidia, blue arrows indicate anatomical regions of the gut, and yellow arrows indicate the horizontal line where signal quantification was performed. Scale bar for confocal microscopy images represents 25 µm and 100 µm for brightfield, respectively. Sample treatment is indicated at the top left corner.

**Figure 5 jof-11-00101-f005:**
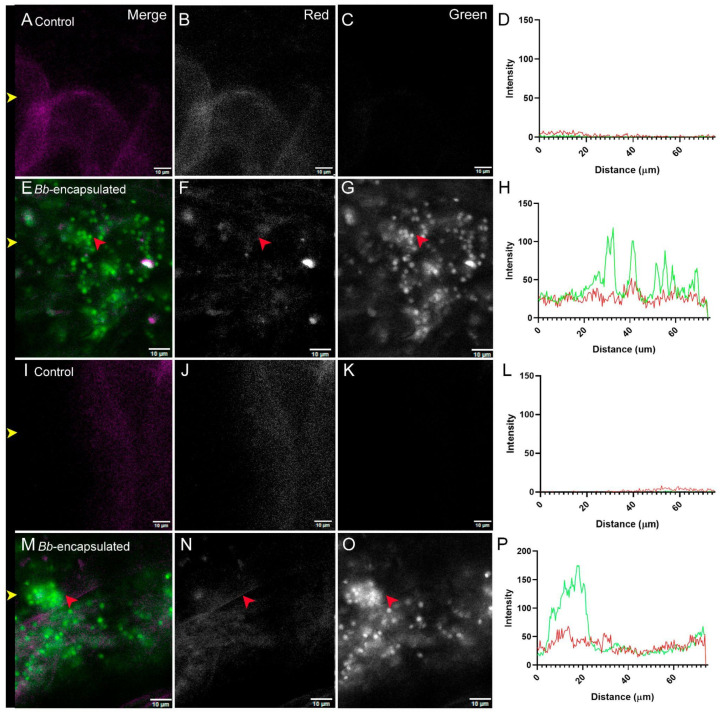
Conidia distribution in *T. castaneum* larvae in the midgut (MG) and ileum (IL) sections of the larval alimentary canal. In control MG (**A**–**D**) and IL (**I**–**L**), no significant green signal is observed or quantified (**D**,**L**).The MG from larvae fed conidia-containing capsules (**E**–**H**) displayed multiple clusters of fungal spores, where individual conidia could be clearly seen (red arrows). Similarly, in the IL from larvae fed conidia-containing capsules (**M**–**P**), clusters of fungal spores (red arrows) and individual conidia are visible. In (**D**,**H**,**L**,**P**), green and red lines indicate signals in the green and red channels, respectively. Channels are indicated in the top-right corner of the top panels. Yellow arrows indicate the horizontal line where signal quantification was measured. Scale bar for confocal microscopy images represents 10 µm. Sample treatment is indicated at the top left corner.

**Figure 6 jof-11-00101-f006:**
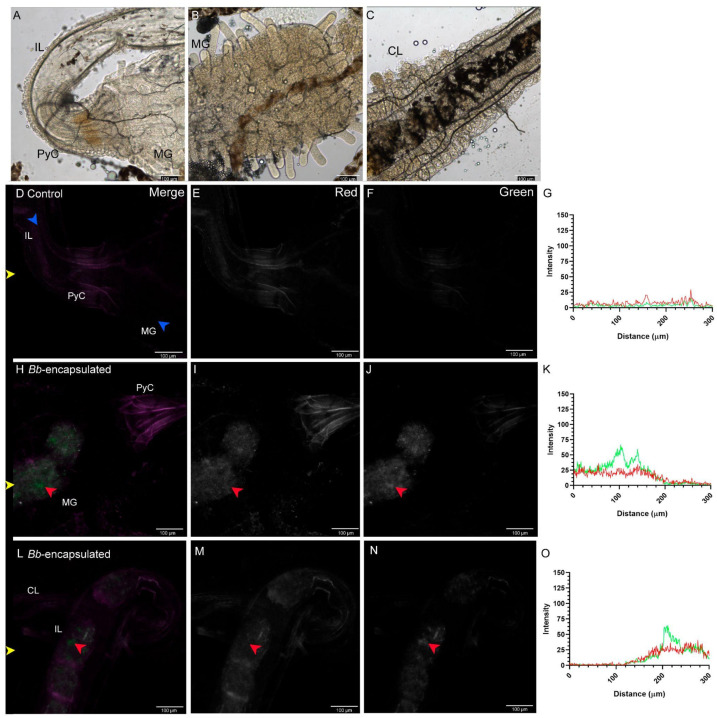
Presence of conidia in the alimentary canal of adult *T. castaneum* fed alginate capsules containing *B. bassiana* conidia. The adult *T. castaneum* alimentary canal has a more complex structure than that of the larval stage: (**A**) ileum (IL) and pyloric chamber (PyC); (**B**) midgut (MG) and (**C**) colon (CL). Control insects did not show significant signal in the green channel (**D**–**G**), whereas insects fed conidia-containing capsules displayed green signal and visible conidia (red arrows) in MG (**H**–**K**), and IL and CL (**L**–**O**). In (**G**,**K**,**O**), green and red lines indicate signals in the green and red channels, respectively. Channels are indicated in the top-right corner of the top panels. Blue arrows indicate anatomical regions of the gut, and yellow arrows indicate the horizontal line where signal quantification was performed. Scale bar for confocal microscopy images represents 100 µm. Sample treatment is indicated at the top left corner.

**Figure 7 jof-11-00101-f007:**
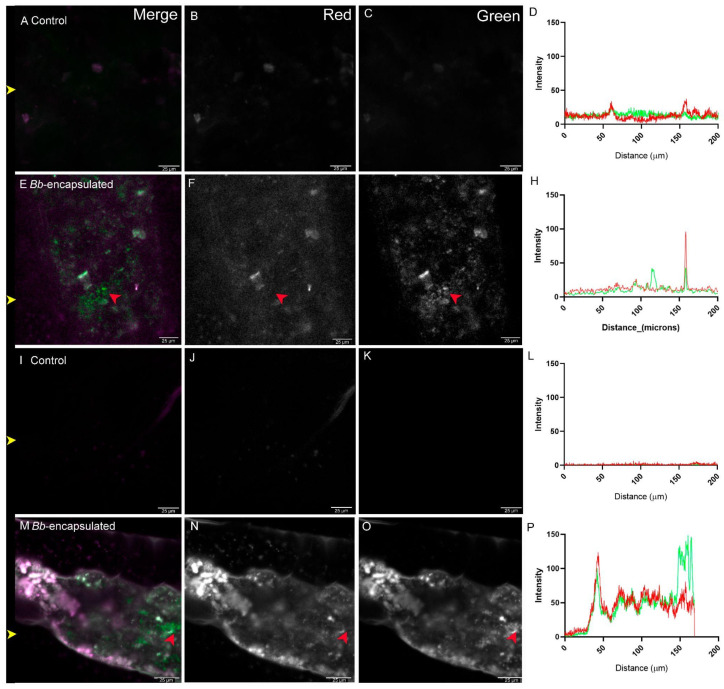
Conidia distribution in the adult *T. castaneum* midgut (MG) and ileum-colon (IL) sections of the alimentary canal. In control insects, no significant green signal is observed in either MG (**A**–**D**) or IL (**I**–**L**), nor is any signal quantified in panels (**D**,**L**). Adults fed conidia-containing capsules (**E**–**H**) displayed multiple clusters of fungal spores in the MG, with individual conidia clearly visible (red arrows). Similarly, in the IL, clusters of fungal spores (red arrows) and individual conidia were observed (**M**–**P**). This is also reflected in the intensity quantification, where peaks in the green channel were detected (**P**). In (**D**,**H**,**L**,**P**), green and red lines indicate signals in the green and red channels, respectively. Channel information is indicated in the top-right corner of the top panels. Red arrows point to conidia, and yellow arrows indicate the horizontal lines where signal quantification was performed. Scale bar for confocal microscopy images represents 25 µm. Sample treatment is indicated at the top left corner.

**Figure 8 jof-11-00101-f008:**
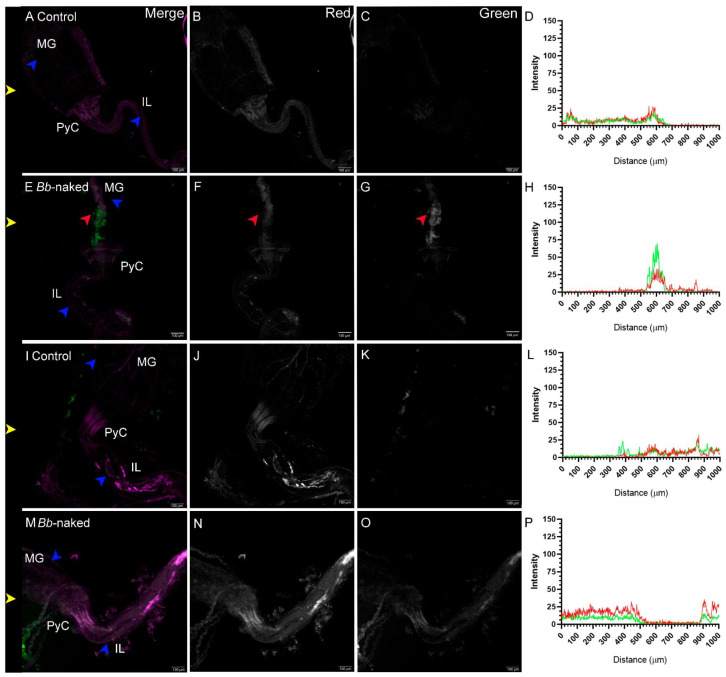
Larvae and adult *T. castaneum* fed with *B. bassiana* naked-conidia mixed into their diet. Control larvae did not show any signal (**A**–**C**) in the green channel (**D**). Larvae fed a conidia-containing diet clearly exhibited conidia in the midgut (MG) (red arrow) (**E**–**G**). This is also reflected in the quantification, where a green signal peak was detected (**H**). For adults, control (**I**–**L**) and fungus-treated samples (**M**–**P**) showed no significant differences. In (**D**,**H**,**L**,**P**), green and red lines indicate signals in the green and red channels, respectively. Channels are indicated in the top-right corner of the top panels. Blue arrows indicate anatomical regions of the gut, and yellow arrows indicate the horizontal line where signal quantification was performed. Scale bar for confocal microscopy images represents 100 µm. Sample treatment is indicated at the top left corner.

## Data Availability

The original contributions presented in this study are included in the article/Appendix A. Further inquiries can be directed to the corresponding authors.

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
