# Peer review of "Visualizing Oral Infection Dynamics of Beauveria bassiana in the Gut of Tribolium castaneum"

_jof, 2025, doi:10.3390/jof11020101_

Round 1

Reviewer 1 Report

This work examines fungal pathogenesis through the penetration of the fungus per os. This approach is quite rare in studies and therefore is interesting. The authors used modern methods for visualization of this process, as well as classical methods. The article is relevant and opens up new directions for research in this issue. However, some clarifications and improvements are needed.

1. The authors focus on the use of hydrogel capsules with fungal conidia. They compare the use of regular dried conidia. However, for a better understanding, a comparison of mortality rates with the use of both dry and encapsulated conidia is required (dried conidia mortality graphs or discussion in the text). What is the mortality with this feeding? Of course, some conidia remain on the surface of insects, but perhaps mortality occurs from the pathogen penetrating through the digestive system and through the cuticle, which is worth discussing.

2. L. 463 "Although conidia ingestion via hydrogel capsules led to mortality in T. castaneum, the underlying mechanisms driving fungal infection following ingestion remain unclear." I would recommend discussing not only intestinal blockage and decreased digestibility, but also consequences and death from toxicosis, since a very large number of fungal propagules enter the host, and perhaps the metabolites released by the fungus could also play a role in mortality.

3. Another interesting point is the presence of conidia in insect feces. If there is data on this, then it is worth providing them, if not, then perhaps this should be taken into account in future studies.

4. What age larvae and adult (after molting ) were used in the experiment? In percutaneous infestation, in addition to the stage of development, the age of the insect is also of great importance in terms of resistance to mycopathogens. Perhaps this is less important in oral infestation, but the age should still be indicated.

5. L. 486 "The insect gut is protected by the peritrophic membrane, a structure composed of chitin microfibrils [22], which may physically hinder fungal adhesion and limit contact with gut tissues." The authors point out that chitin, which is part of the peritrophic membrane, can protect against penetration of the fungus, but during percutaneous infection the fungus has adapted to penetrate through chitinized surfaces. Perhaps there is a more complex interaction here and it is worth explaining this, rather than simply referring to the presence of chitin.

6. 2.5. "Colony-forming unit counts. …The alimentary canals were placed in physiological solution, vortexed for 1 minute, diluted to a final volume of 1 mL, and plated onto PDA." Was shaking the alimentary canals in physiological solution sufficient to release the spores? Perhaps, homogenization was necessary. Was only the solution or also intestinal tissue used for the plated?

7. Figure S4. In the caption it is should be indicated that this is for encapsulated conidia.

8. There is an article (https://doi.org/10.1371/journal.pone.0081686), though done on mosquitoes using the fungus Metarhizium, which shows the presence of fungal spores in the intestinal lumen and the rapid death of larvae without spore germination. If you think it is appropriate, you can add her the discussion.

Author Response

Response to Reviewer 1 Comments

1. Summary

Thank you very much for taking the time to review this manuscript. Please find the detailed responses below and the corresponding revisions/corrections highlighted in yellow in the re-submitted files.

2. Questions for General Evaluation

Reviewer’s Evaluation

Response and Revisions

Does the introduction provide sufficient background and include all relevant references?

Yes

Are all the cited references relevant to the research?

Yes

Some references were added or switched following reviewers’ suggestions.

Is the research design appropriate?

Yes

Are the methods adequately described?

Yes

Are the results clearly presented?

Yes

Are the conclusions supported by the results?

Yes

3. Point-by-point response to Comments and Suggestions for Authors

Comments 1: The authors focus on the use of hydrogel capsules with fungal conidia. They compare the use of regular dried conidia. However, for a better understanding, a comparison of mortality rates with the use of both dry and encapsulated conidia is required (dried conidia mortality graphs or discussion in the text). What is the mortality with this feeding? Of course, some conidia remain on the surface of insects, but perhaps mortality occurs from the pathogen penetrating through the digestive system and through the cuticle, which is worth discussing.

Response 1: Thank you for your insightful comment. We appreciate your suggestion to include a comparative analysis of mortality rates for both naked dried conidia and encapsulated conidia. In response, we have incorporated panel B in Figure 3, which presents the cumulative mortality for naked dried conidia (page 8, line 324 of the manuscript). Furthermore, we have provided a detailed comparative discussion of these findings with the cumulative mortality for encapsulated conidia, as shown in Figure 3, panel A (page 7, section 3.2, lines 311–320). These additions and revisions have been highlighted in yellow for your convenience.

Comments 2: L. 463 "Although conidia ingestion via hydrogel capsules led to mortality in T. castaneum, the underlying mechanisms driving fungal infection following ingestion remain unclear." I would recommend discussing not only intestinal blockage and decreased digestibility, but also consequences and death from toxicosis, since a very large number of fungal propagules enter the host, and perhaps the metabolites released by the fungus could also play a role in mortality.

Response 2: Thank you for your valuable suggestion. We fully agree with your recommendation to expand the discussion to include the potential role of secondary metabolite toxicosis in mortality. Accordingly, we have revised the discussion section by adding a paragraph that highlights this aspect, including the possible contribution of fungal metabolites to mortality in addition to intestinal blockage and decreased digestibility. This insightful comment has significantly enriched our discussion. The changes are reflected in Section 4 of the manuscript, on pages 14–15, lines 525–553, and are highlighted in yellow for your convenience.

Comments 3: Another interesting point is the presence of conidia in insect feces. If there is data on this, then it is worth providing them, if not, then perhaps this should be taken into account in future studies.

Response 3: Thank you for highlighting this interesting and important point. Regrettably, we do not currently have data on the presence of conidia in insect feces, but we agree that this is a valuable area for future investigation. Notably, in our 2019 review paper (Mannino et al., 2019; DOI: 10.3390/jof5020033), we discussed evidence suggesting the presence of ungerminated conidia in insect feces, which underscores the relevance of this phenomenon. We appreciate your suggestion and will consider it as a focus for future studies.   

Comments 4: What age larvae and adult (after molting ) were used in the experiment? In percutaneous infestation, in addition to the stage of development, the age of the insect is also of great importance in terms of resistance to mycopathogens. Perhaps this is less important in oral infestation, but the age should still be indicated.

Response 4: Thank you for highlighting the importance of specifying the age of the insects used in the experiment. We apologize for the omission and have now included this information in the Materials and Methods section (Section 2.2, page 3, lines 132–134). The age of the larvae and adults (post-molting) is now clearly indicated to address your concern. These changes are highlighted in yellow in the manuscript for your convenience.

Comments 5: L. 486 "The insect gut is protected by the peritrophic membrane, a structure composed of chitin microfibrils [22], which may physically hinder fungal adhesion and limit contact with gut tissues." The authors point out that chitin, which is part of the peritrophic membrane, can protect against penetration of the fungus, but during percutaneous infection the fungus has adapted to penetrate through chitinized surfaces. Perhaps there is a more complex interaction here and it is worth explaining this, rather than simply referring to the presence of chitin.

Response 5: Thank you for this insightful comment. We appreciate your suggestion to provide a more nuanced explanation of the interaction between fungal penetration and the chitinized structure of the peritrophic membrane. In response, we have expanded the discussion to address the potential complexity of these interactions, beyond merely referencing the presence of chitin. The revised content is included in the Discussion section (Section 4, page 15, lines 554–565 and 570–573). These changes, which we believe significantly enhance the depth of the discussion, are highlighted in yellow in the manuscript for your convenience.

Comments 6: 2.5. "Colony-forming unit counts. …The alimentary canals were placed in physiological solution, vortexed for 1 minute, diluted to a final volume of 1 mL, and plated onto PDA." Was shaking the alimentary canals in physiological solution sufficient to release the spores? Perhaps, homogenization was necessary. Was only the solution or also intestinal tissue used for the plated?

Response 6: Thank you for highlighting the need for a more detailed description of our method. We have clarified the procedure used for processing the alimentary canal in the Materials and Methods section (Section 2.5, page 5, lines 221–223), with changes highlighted in yellow. The primary objective of this experiment was to confirm the presence of conidia within the alimentary canal, and we applied the same protocol across all samples, including technical and biological replicates. While the solution obtained after vortexing was plated, the intestinal tissue itself was not plated separately. We agree that homogenization or plating the tissue directly could provide additional insights, and we appreciate this suggestion for future studies. To account for potential limitations in conidia quantification through this method, we also performed direct counts using confocal microscopy images.

Comments 7: Figure S4. In the caption it is should be indicated that this is for encapsulated conidia.

Response 7: Thank you for pointing out the unclear labeling in the caption of Figure S4. We have addressed this by explicitly indicating that the figure pertains to encapsulated conidia. The modification has been made in the Supplementary Material file (page 5) and is highlighted in yellow for your convenience.

Comments 8: There is an article (https://doi.org/10.1371/journal.pone.0081686), though done on mosquitoes using the fungus Metarhizium, which shows the presence of fungal spores in the intestinal lumen and the rapid death of larvae without spore germination. If you think it is appropriate, you can add her the discussion.

Response 8: Thank you for this insightful suggestion and for directing us to the relevant publication. We have incorporated content from the recommended article into the discussion to provide additional context and support. The changes have been added to the Discussion section (Section 4, pages 491–500 and 525–528) and are highlighted in yellow for your convenience.

4. Response to Comments on the Quality of English Language

Point 1: not applicable.

Response 1:    (in red)

5. Additional clarifications

Other reviewers suggestion can be found on the manuscript highlighted in green and blue.

Reviewer 2 Report

In the current study by Preisegger et al., following oral exposure of the red flour beetle to encapsulated Beauveria bassiana conidia formulation, with the aid of confocal microscopy, the authors examined the progression and dynamics of conidia in the gut of the insect.

A study of this kind, elucidating the unique mode of infection of important insect pathogenic fungus, such as B. bassiana is important. I argue that the work is innovative and merits publication in JOF. I do not have any major issues with the study or its contents, except for minor issues with the manuscript. The authors are encouraged to make minor changes to the manuscript, as highlighted in the detail comments section below.

The authors need to highlight the innovative contributions of this paper. Other than just concluding that “This disruption may weaken the host, increasing its susceptibility to infections and, ultimately, leading to death.”, a strong statement on why the paper is important (i.e. what does it add to science) could conclude the abstract.

L42-44 – …relating to the ability of fungi to release enzymes and toxins that compromise the immune systems of target insects, the authors wrote “Several studies have identified specific fungal genes involved in these processes although the molecular mechanisms involved remain still under investigation.” One would expect them to reference a few of the ‘Several studies…” but the ref [4,5] – one is an editorial, the other is a review article. Please, cite original articles here.

L89 – Indicate the origin/source of the fungus.

L93 – essays? - ‘assays’ you mean?

L166 – Ensure that the correct symbol for ‘Celsius’ is used and be consistent all over the manuscript. For instance, 45°C was used in L166, 4oC in L173. Express as ‘45 °C’ and correct all over the manuscript.

L169, 178, 186, 198, 211 – Revise as “26 °C” Check all over the manuscript.

L87-234 - The study was well conducted and adequately replicated. I confirm that the experimental procedures were well described and can easily be duplicated.

L260 – The scales on the Fig. 1. is not clear. Use a bigger font. Do the same for figures 2 and 4.

L308 – Is the caption of Figure 3 describing the figure above? I am wondering what color is ‘light green’ or ‘dark green’ in the figure. The color seems grey and black, please recheck.

L379 – Fig. 6. Use a bigger font for the scales. Do the same for other Figs. 7 and 8

L591-592 – Please, re-check this reference.

Author Response

Response to Reviewer 2 Comments

1. Summary

Thank you very much for taking the time to review this manuscript. Please find the detailed responses below and the corresponding revisions/corrections highlighted in green in the re-submitted files.

2. Questions for General Evaluation

Reviewer’s Evaluation

Response and Revisions

Does the introduction provide sufficient background and include all relevant references?

Yes

[Please give your response if necessary. Or you can also give your corresponding response in the point-by-point response letter. The same as below]

Are all the cited references relevant to the research?

Yes

Is the research design appropriate?

Yes

Are the methods adequately described?

Yes

Are the results clearly presented?

Can be improved, figure scales

Are the conclusions supported by the results?

Yes

3. Point-by-point response to Comments and Suggestions for Authors

Comments 1: The authors need to highlight the innovative contributions of this paper. Other than just concluding that “This disruption may weaken the host, increasing its susceptibility to infections and, ultimately, leading to death.”, a strong statement on why the paper is important (i.e. what does it add to science) could conclude the abstract.  

Response 1: Thank you for your valuable suggestion. To strengthen the impact of the abstract and better highlight the significance of our work, we have revised the conclusion to emphasize the innovative contributions of this research. These modifications can be found in the Abstract section of the manuscript (page 1, lines 36–39) and are highlighted in green for your convenience.

Comments 2: L42-44 – …relating to the ability of fungi to release enzymes and toxins that compromise the immune systems of target insects, the authors wrote “Several studies have identified specific fungal genes involved in these processes although the molecular mechanisms involved remain still under investigation.” One would expect them to reference a few of the ‘Several studies…” but the ref [4,5] – one is an editorial, the other is a review article. Please, cite original articles here.  

Response 2: Thank you for your valuable comment. We agree that it is important to cite original research articles in this context. To address this, we have updated the references by including original studies related to the topic, replacing the previous citations with more appropriate sources. These changes are reflected in the Introduction section (page 2, line 48) and the References section (page 17, lines 674–683), and are highlighted in green for your convenience.

Comments 3: L89 – Indicate the origin/source of the fungus.

Response 3: Thank you for pointing out the missing information. We have now included the source of the fungal material in the Materials and Methods section (Section 2.1, page 2, lines 95–96). The modification is highlighted in green for your convenience.

Comments 4: L93 – essays? - ‘assays’ you mean?

Response 4: Thank you for pointing out the typo. We have corrected the misspelled word to "assays." The modification can be found in the Materials and Methods section (Section 2.1, page 3, line 100) and is highlighted in green for your convenience.

Comments 5: L166 – Ensure that the correct symbol for ‘Celsius’ is used and be consistent all over the manuscript. For instance, 45°C was used in L166, 4oC in L173. Express as ‘45 °C’ and correct all over the manuscript.

Response 5: Thank you for pointing out the inconsistency in the use of the Celsius symbol. We have corrected all instances throughout the manuscript to ensure consistency. The necessary corrections have been made in the following lines: 97, 120, 123, 177, 180, 185, 190, 198, 202, 210, and 223. These changes are highlighted in blue for your convenience.

Comments 6: L169, 178, 186, 198, 211 – Revise as “26 °C” Check all over the manuscript.

Response 6: We appreciate your comment. In accordance with your suggestion, we have revised the instances of “26 °C” in the manuscript. These changes are also highlighted in blue for your convenience.

Comments 7: L87-234 - The study was well conducted and adequately replicated. I confirm that the experimental procedures were well described and can easily be duplicated.

Response 7: Thank you very much for your positive feedback. We greatly appreciate your recognition of the study's design and methodology. We are pleased to know that the experimental procedures were clearly described and can be easily duplicated. Your comments are highly encouraging and motivate us to continue our work with confidence.

Comments 8: L260 – The scales on the Fig. 1. is not clear. Use a bigger font. Do the same for figures 2 and 4

Response 8: Thank you for pointing out the issue with the scales in Figure 1. We have increased the font size for the scales in Figures 1 (page 6, line 272), 2 (page 7, line 292), and 4 (page 8, line 343), and have adjusted the captions accordingly to accommodate these changes.

Comments 9: L308 – Is the caption of Figure 3 describing the figure above? I am wondering what color is ‘light green’ or ‘dark green’ in the figure. The color seems grey and black, please recheck.

Response 9: Thank you for your careful observation. You are correct that the color description in the caption of Figure 3 was inconsistent. We have updated the caption to accurately reflect the colors used in the figure. The modification can be found in the Results section (Section 3.2, Figure 3 caption, page 8, line 328).

Comments 10: L379 – Fig. 6. Use a bigger font for the scales. Do the same for other Figs. 7 and 8

Response 10: Thank you for pointing out the issue with the scales in Figures 6, 7 and 8. We have increased the font size for the scales in Figures 6 (page 11, line 411), 7 (page 12, line 444), and 8 (page 13, line 467), and have adjusted the captions accordingly to accommodate these changes.

Comments 11: L591-592 – Please, re-check this reference.

Response 11: Thank you for pointing that out. In line with your suggestion in comments 2, we have removed the incorrect reference and replaced it with the appropriate ones in the References section.

4. Response to Comments on the Quality of English Language

Point 1: not applicable

Response 1:    (in red)

5. Additional clarifications

Other reviewers’ suggestions and corrections are highlighted in yellow and blue.

Reviewer 3 Report

Dear authors,

congratulations, The results obtained in this study are more than robust, but I strongly suggest improving their presentation to make it more intuitive for the reader. see suggestion inside of document attachement

see detail inside of document

Author Response

Response to Reviewer 3 Comments

1. Summary

Thank you very much for taking the time to review this manuscript. Please find the detailed responses below and the corresponding revisions/corrections highlighted in blue in the re-submitted files.

2. Questions for General Evaluation

Reviewer’s Evaluation

Response and Revisions

Does the introduction provide sufficient background and include all relevant references?

Yes

Are all the cited references relevant to the research?

Yes

Is the research design appropriate?

Yes

Are the methods adequately described?

Yes

Are the results clearly presented?

Can be improved

Are the conclusions supported by the results?

Yes

3. Point-by-point response to Comments and Suggestions for Authors

Comments 1: Celsius sign inconsistency  

Response 1: Thank you for pointing out the inconsistency in the use of the Celsius symbol. We have corrected all instances throughout the manuscript to ensure consistency. The necessary corrections have been made in the following lines: 97, 120, 123, 177, 180, 185, 190, 198, 202, 210, and 223. These changes are highlighted in blue for your convenience.

Comments 2: Your first result goes into the supplementary material. ??

Response 2: Thank you for your comment. We understand your concern regarding the placement of our first result in the supplementary material. However, due to the extensive nature of the data, we felt that the simple encapsulated size measurements, while important, were less central to the primary focus of the paper. For this reason, we chose to include these measurements in the supplementary material.

Comments 3: Present/write your results in a sequential manner as the figure 2 (or change the order of the figures or the writing of the results), please

Response 3: Thank you for your suggestion. We agree with the need for a more sequential presentation of the results. We have revised the description of the figures to follow a more logical order. The changes can be found in the Results section (Section 3.1, pages 6–7, lines 285–291), and are highlighted in blue for your convenience.

Comments 4: A????, A is alimentary canal. Check please

Response 4: Thank you for pointing out the discrepancy. We have corrected the figure caption to accurately reflect the information displayed. The correction can be found in the Results section (Figure 4 caption, page 8, lines 348–350), and the changes are highlighted in blue for your convenience.

Comments 5: Present/write your results in a sequential manner as the figure 5 (or change the order of the figures or the writing of the results), please. Highlight the letter that represents the figure in the caption   

Response 5: Thank you for your suggestion. We agree that the results should be presented sequentially, and we have revised the description to align with Figure 5. The changes can be found in the Results section (Section 3.3, page 9, lines 375–384), and are marked in blue for clarity.   

Comments 6: Clarify Control and Treatment on figures

Response 6: Thank you for your feedback. We have added clear labels for control and treatment conditions in all figures to ensure consistency and ease of understanding throughout the Results section. 

Comments 7: Describe Figure 8 accurately

Response 7: Thank you for your comment. We have revised the description of Figure 8 to ensure it is accurate and complete. The changes can be found in the Results section (Section 3.3, page 12, lines 459–460), and are marked in blue for your convenience.

4. Response to Comments on the Quality of English Language

Point 1: not applicable

Response 1:    (in red)

5. Additional clarifications

Other reviewers suggestions are also marked in the manuscript in yellow and green